# Unified Music-Language Model for Symbolic and Waveform Integration

## Abstract

Music is a unique and essential modality constituting human life, presenting challenges for multimodal advances due to its complex structure and intricate details. Recent Music Language Models (MuLMs) facilitate music understanding and generation by leveraging the inherent knowledge and reasoning capabilities of pre-trained Language Models (LMs), yet they overlook the complementary benefits of different music representations. To this end, we propose a unified music language model, named UniMuLM, form the existing approach of using a single representation to multiple music representations. Concerning the unification, we address the challenges of missing modalities and unstable training to adapt different scenarios. Specifically, we integrate symbolic, waveform music, and textual instructions into an LM and design a bar-level tokenizer to explore the fine-grained correlations between different modalities. Moreover, we propose a multi-stage training strategy to progressively enhance this synergy. Trained on open-source datasets, UniMuLM demonstrates superior performance compared to SOTA methods across five music tasks, evaluated on nine benchmark datasets. The demo examples can be accessed via `https://anonymous-2024101.github.io`.

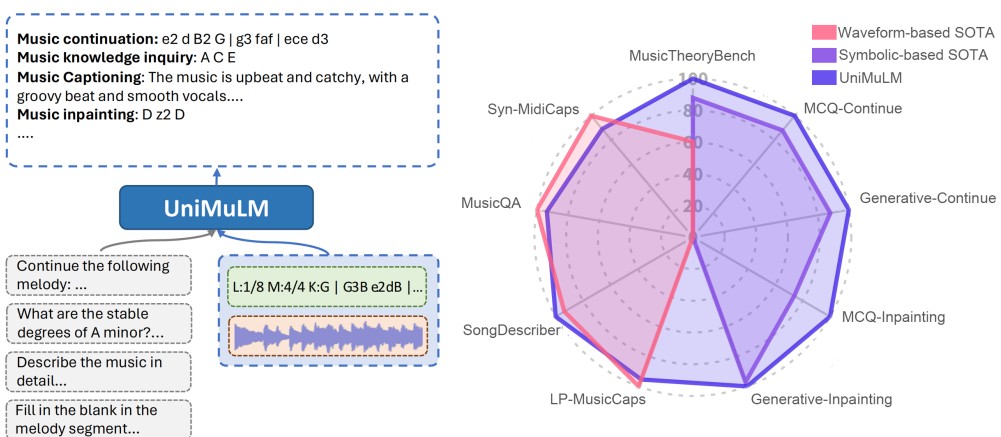

Figure 1: UniMuLM (our method) is capable of handle a range of music tasks (left), including music continuation, music knowledge inquiry, and more by taking symbolic, waveform music, and textual instructions as input. UniMuLM achieves SOTA results (right) on most of music tasks compared to previous waveform-based and symbolic-based approaches (the results are normalized such that the maximum score of all models is 100% for each task).

## 1 Introduction

Language Models (LMs) have recently made remarkable progress in various linguistic tasks (Brown et al., 2020; Team, 2024). By leveraging extensive pre-trained corpora and demonstrating impressive reasoning capabilities, LMs show significant potential for understanding multimodal content, motivating researchers to explore Multimodal Language Models (MLMs) (Liu et al., 2023; Alayrac et al., 2022; Wang et al., 2023). Among the diverse modalities, music stands out due to its unique

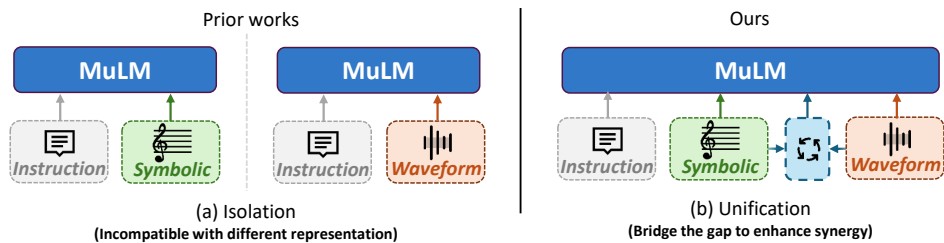

Figure 2: Paradigm comparison between our method and prior works in multiple music representations integration. (a) Prior works focus on utilizing either symbolic or waveform representations, typically in isolation. (b) In contrast, our approach not only incorporates both representations but also introduces a unification mechanism that bridges the gap and enhances their synergy.

blend of rhythm, melody, harmony, and lyrics, capable of evoking emotion. This has sparked significant interest in developing Music Language Models (MuLMs), which aim to address various music-related tasks, such as music question answering, inpainting, and continuation (Chu et al., 2023; Liu et al., 2024; Agostinelli et al., 2023; Deng et al., 2024; Tang et al., 2024) with a single model.

A pivotal obstacle preventing MuLMs from becoming true experts in music lies in the commonly adopted approach of treating music representations as either symbolic notations or raw waveforms, with most models designed to handle only one of these forms. Yet, for music experts, notation and performance are unified — *even Beethoven's deafness could not stop him from hearing music in his mind*. For example, MidiCaps (Melechovský et al., 2024) extracts meta-information like tonality and rhythm from MIDI but relies on waveform models for semantic information after synthesis. This highlights how current research remains fragmented, failing to capitalize on the potential of integrating both forms. The root cause of this limitation is the temporal scale inconsistency between music representations. Symbolic music uses uneven time divisions determined by note durations, while raw waveform signals are represented at a much higher sampling frequency. Despite various efforts to encode waveforms differently, such as using CNNs for sampling (van den Oord et al., 2016), VAEs for discrete encoding (Zeghidour et al., 2022), self-supervised representation learning (Li et al., 2022), or aligning audio with text for global representations (Elizalde et al., 2023), none have effectively bridged the gap to form a consistent representation with symbolic music. This inconsistency necessitates models to learn each representation in isolation, preventing the development of a unified understanding of music and, thus, the ability to leverage the complementary strengths of both representations.

However, training a model that unifies these representations for diverse music tasks is inherently complex. Scenarios where all three modalities (*i.e.,* symbolic, waveform music, and textual instruction) appear together are rare, while certain tasks require two of them (Ji et al., 2020). This presents two key challenges for our research: first, exploring how the model can still benefit when one modality is absent; second, ensuring the stability of multitask training so that tasks enhance rather than hinder each other.

To this end, we introduce UniMuLM, a Unified Music-Language Model, which is not only compatible for both symbolic music and waveforms but also unifies them at a bar level to achieve fine-grained, mutually reinforced representations. We employ a multi-stage approach to train UniMuLM, which consists of three stages. First, we start by leveraging music knowledge and symbolic music datasets (*e.g.,* MusicPile (Yuan et al., 2024) and MelodyHub (Wu et al., 2024)) to inject music knowledge and warm up the LM base model, such as Llama3 (Team, 2024). Next, we train a bar-level tokenizer using paired symbolic music and waveforms to pre-align and bridge the gap between music representations. Lastly, we apply LoRA-tuning (Hu et al., 2022) to the LM and update adapters for diverse musical representations across all downstream tasks using different datasets, including MusicCaps (Agostinelli et al., 2023), Song-Describer (Manco et al., 2023), MidiCaps (Melechovský et al., 2024), and MusicQA (Liu et al., 2024).

Our contributions are threefold. (1) We emphasize the often-overlooked complementarity of music representations and propose UniMuLM, a unified framework that integrates symbolic, wave-

form music, and textual instructions. (2) We explore the fine-grained correlation between different music representations at the bar level with explicit tokenization in UniMuLM. (3) We benchmark UniMuLM using both quantitative and qualitative metrics across 9 tasks. The superior performance on these tasks demonstrates the efficacy of unifying different music representations and the soundness of our design.

## 2  RELATED WORK

We briefly review the related works from two aspects: Music Encoding, where we explore symbolic and waveform music representations; and Music Language Models, which expand language models to incorporate music.

### 2.1  MUSIC ENCODING

Music encompasses symbolic notation, lyrics, and waveforms produced by instruments and vocalists (Ji et al., 2020), with deep learning-based research generally classifying these representations into two categories: symbolic and waveform.

**Waveform music** consists of one-dimensional signals sampled at high frequencies. While some models directly process raw waveforms (van den Oord et al., 2016; Baevski et al., 2020), a more commonly adopted approach transforms waveforms into spectrograms using the Fourier Transform (FT) (Gong et al., 2022), which provides a richer representation of audio information. Additionally, some models treat waveforms or spectrograms as images and leverage diffusion frameworks for music generation (Forsgren & Martiros, 2022). Currently, the most popular method involves converting music into discrete tokens using Variational Autoencoders (VAEs) (Défossez et al., 2023) for music generation and text-aligned tasks (Dhariwal et al., 2020; Castellon et al., 2021).

**Symbolic music** has garnered widespread attention in the deep learning community due to its representation of discrete notations. MIDI carries real-time performance and control data for specific notes and is widely used by musicians and producers globally (Ji et al., 2020). Beyond direct process with raw MIDI (Hadjeres et al., 2017; Lu et al., 2023; Zeng et al., 2021; Huang et al., 2019; Dong et al., 2018), many derivative representations of MIDI have emerged, aiming to reduce sequence length, improve readability, increase information density, and integrate multi-track information. Examples of such representations include REMI (Huang & Yang, 2020), OctupleMIDI (Zeng et al., 2021), Humdrum (Cherla et al., 2015), and CompoundWord (Hsiao et al., 2021). Recent studies highlight ABC notation's data efficiency, alignment with human compositional practices, and extensive community support, making it a preferred format for MuLM (Yuan et al., 2024; Qu et al., 2024; Wu et al., 2024). UniMuLM follows the ABC choice of these works.

### 2.2  MUSIC LANGUAGE MODEL

MuLMs are frameworks that model music understanding or generation as sequential token prediction. MuLMs adapt techniques from other MultiModal-LMs (Alayrac et al., 2022; Driess et al., 2023; Wang et al., 2023; Liu et al., 2023; Yang et al., 2023; Kong et al., 2024), which have rapidly evolved due to the knowledge retention, reasoning, and instruction-following abilities of language models. A key factor determining the performance of MuLMs is how musical representations are processed and input into the model. And current approaches are basically sorted into two strands: waveform- and symbolic-based methods.

**Waveform**-based MuLM encompasses various methods for encoding audio into LM-compatible inputs, with two main approaches: encoding audio into discrete acoustic tokens or providing audio features through a trained adapter. The former approach, as explored by Jukebox (Dhariwal et al., 2020), utilizes a VAE to encode audio into acoustic tokens for music reconstruction, followed by AudioLM (Borsos et al., 2023), VampNet (García et al., 2023), and VALL-E2 (Chen et al., 2024), which leverage RVQ-VAEs (Zeghidour et al., 2022; Défossez et al., 2023). While these methods are able to reconstruct finer audio features, they typically require larger-scale training. In contrast, for the adapter approach, researchers utilize the waveform features extracted by MERT (Li et al., 2024) and CLAP (Elizalde et al., 2023). For example, Deng et al. (2024) and Tang et al. (2024) use the

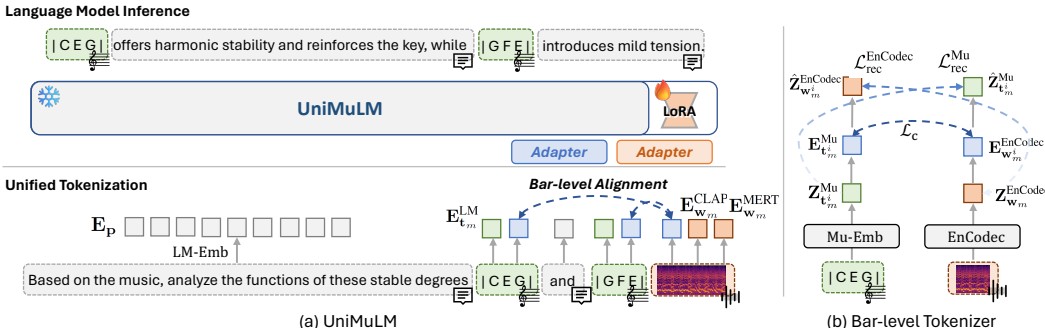

(a) UniMuLM       (b) Bar-level Tokenizer

Figure 3: The overall framework of UniMuLM (a), which consists of a unified tokenization and a base language model. Specifically, UniMuLM takes textual instructions, symbolic and waveform representations as the input, where different music representations are aligned at the bar level, and the base language model is fine-tuned to adapt to various music tasks. Notably, UniMuLM employs a bar-level tokenizer (b), which is trained via contrastive loss and corss-reconstruction loss, to explicitly model the alignment between symbolic and waveform information.

embeddings encoded by adapters as direct inputs to the LM decoder, while others (Liu et al., 2024) adopt a cross-attention mechanism.

**Symbolic**-based MuLM employs two primary tokenization strategies: either creating a pre-defined or custom-trained tokenizer or using a pre-trained LM's text tokenizer. For the first approach, Oore et al. (2020) use on-off representations, the PopMAG (Ren et al., 2020) tokenizer focuses on duration-based information, while MuPT (Qu et al., 2024) customizes a BPE (Fradet et al., 2023) tokenizer specifically for ABC notation. These methods require training from scratch. In contrast, the second approach involves directly using a pre-trained LM's text tokenizer, treating notations as second languages. Notable examples include ChatMusician (Yuan et al., 2024), which utilizes the LLaMA2 tokenizer, allowing it to leverage the world knowledge and reasoning capabilities of large pre-trained language models.

UniMuLM is compatible with both waveform and ABC notation input. The waveform is processed through CLAP and MERT, with its features passed through an adapter as embeddings into the initial layers of LLaMA. ABC is tokenized by the frozen LLaMA3 tokenizer as well as processed by a bar-level tokenizer, resulting in a dual-representation.

## 3 PROBLEM FORMULATION

Given a piece of music $m = \{\mathbf{t}_m, \mathbf{w}_m\}$, where $\mathbf{t}_m$ and $\mathbf{w}_m$ denote the symbolic and waveform representations of $m$, and the prompt $\mathbf{p}$, which indicates the specific task, *e.g., "Describe the music in detail"* for music captioning, our objective is to obtain a MuLM that can accordingly generate the desired answer $\mathbf{a}$. Specifically, $\mathbf{t}_m \in \mathcal{T}^{l_m}$ represents the textual ABC notations consisting of $l_m$ text tokens from the token set $\mathcal{T}$, while $\mathbf{w}_m \in \mathbb{R}^{s \cdot r_s}$ is a sequence of sampling points with a duration of $s$ and a sampling rate of $r_s$. $\mathbf{p} \in \mathcal{T}^{l_p}$ and $\mathbf{a} \in \mathcal{T}^{l_a}$ are similarly sequences of text tokens that define the downstream tasks and ground-truth answers. According to the general definition of a language model, we can frame this as an autoregressive estimation: $P(\mathbf{a}_i | \mathbf{p}, \mathbf{t}_m, \mathbf{w}_m, \mathbf{a}_{1:i-1})$.

## 4 UNIFIED MUSIC-LANGUAGE MODEL

UniMuLM consists of a unified tokenization to handle different music representations and textual instructions, with a language model serving as the backbone, as shown in Figure 3 (a). Moreover, a multi-stage training strategy is introduced to progressively optimize the parameters.

### 4.1 UNIFIED TOKENIZATION

To address the issue of incompatibility between different music representations in existing models and to achieve a unified representation that leverages complementary information, we tokenize the

different data formats using corresponding tokenization methods. We then introduce a mechanism of bar-level tokenization to align low-level correspondences across these music representations.

### 4.1.1 SINGLE-MODAL TOKENIZATION

**Instruction Tokenization.** We utilize the pre-defined embedding table from the LM to encode instruction tokens (*e.g.*, , prompt $\mathbf{p}$) via the look-up function LM-Emb : $\mathcal{T} \to \mathbb{R}^d$, where $d$ denotes the dimensionality of the textual embedding. Specifically, $\mathbf{E_p} = \text{LM-Emb}(\mathbf{p}) \in \mathbb{R}^{l_p \times d}$ represents the transformed embeddings, having the same length $l_p$ as the input tokens.

**Symbolic Tokenization.** We encode the symbolic notations into (1) language-level representations for linguistic comprehension by the LM backbone, and (2) music-level representations via a novel tokenizer specifically designed for musical understanding. For language-level representation, we follow the instruction tokenization as formulated: $\mathbf{E}_{\mathbf{t}_m}^{\text{LM}} = \text{LM-Emb}(\mathbf{t}_m) \in \mathbb{R}^{l_m \times d}$. Inspired by Wu et al. (2024), we obtain bar-level representations $\mathbf{Z}_{\mathbf{bar}_i}^{\text{Mu}} = \text{Mu-Emb}(\mathbf{bar}_i) \in \mathbb{R}^{l_m \times d_{\text{mu}}}$ via a new tokenizer Mu-Emb : $\mathcal{T} \to \mathbb{R}^{d_{\text{mu}}}$, which features music-specialized embedding table.

**Waveform Tokenization.** We encode the waveform into (1) high-level representations using CLAP (Elizalde et al., 2023) and MERT (Li et al., 2024), which are specifically designed for music retrieval tasks while capturing global semantics and contextual information, and (2) low-level representations through EnCodec (Roberts et al., 2018), which quantizes continuous music signals into discrete codes to preserve extensive acoustic details. For high-level tokenization, we obtain dense features by applying $\mathbf{E}_{\mathbf{w}_m}^{\text{CLAP}} = \text{CLAP}(\mathbf{w}_m) \in \mathbb{R}^{1 \times d_c}$ and $\mathbf{E}_{\mathbf{w}_m}^{\text{MERT}} = \text{MERT}(\mathbf{w}_m) \in \mathbb{R}^{1 \times d_m}$ from off-the-shelf encoders for efficiency, where $d_c$ and $d_m$ represent the latent sizes of the corresponding representations. For low-level tokenization, the lengthy waveform $\mathbf{w_m}$ is compressed into $\mathbf{Z}_{\mathbf{w}_m}^{\text{EnCodec}} = \text{EnCodec}(\mathbf{w_m}) \in \mathbb{R}^{s \cdot r_c \times d_e}$, where $r_c \ll r_s$ is the frame rate and $d_e$ is the latent size of the codes used in Residual Vector Quantization (RVQ) (Zeghidour et al., 2022).

### 4.1.2 BAR-LEVEL CROSS-MODAL TOKENIZATION

Despite the temporal misalignment between discrete symbolic notations and continuous waveforms, both can be divided into associated segments via bars. For instance, Figure 4 shows an example where the ABC notation is aligned with waveform segments, with bar boundaries explicitly indicated.

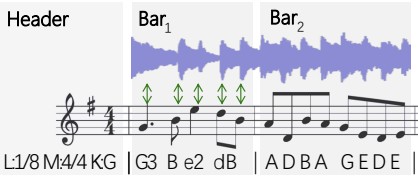

Figure 4: Illustraion of the mapping between the waveform (top) and ABC notations (bottom) in the bar level, where the temporal alignment is highlighted with bidirectional green arrows.

Hence, we propose a bar-level tokenizer to explicitly construct the correspondence between symbolic and waveform music in a fine-grained manner, as shown in Figure 3 (b). Specifically, we retain the header section and split the ABC tune into bars, $\mathbf{t}_m = \{\mathbf{t}_m^0, \mathbf{t}_m^1, \cdots, \mathbf{t}_m^n\}$, where $\mathbf{t}_m^0$ represents the header section, and $n$ represents the number of bars. Then, we synthesize the corresponding waveform using random instruments to generate paired ABC-waveform data: $b_m^i = \mathbf{t}_m^i, \mathbf{w}_m^i, i \in [0, n]$. Following the aforementioned tokenization, we encode each bar as $\mathbf{Z}_{\mathbf{t}_m^i}^{\text{Mu}}$ and $\mathbf{Z}_{\mathbf{w}_m^i}^{\text{EnCodec}}$ using Mu-Emb and EnCodec, respectively.

The model follows an autoencoder structure and consists of four components: Symbolic-Encoder, Wave-Encoder, Symbolic-Decoder, and Wave-Decoder, each built from self-attention layers and multi-layer perceptrons (MLPs). The encoders are enhanced with positional embeddings and LayerNorm to ensure stable training and effective sequence processing, with more details provided in the Appendix. The intermediate embeddings for the ABC and waveform inputs are computed as $\mathbf{E}_{\mathbf{t}_m^i}^{\text{Mu}} = \text{Symbolic-Encoder}(\mathbf{Z}_{\mathbf{t}_m^i}^{\text{Mu}}) \in \mathbb{R}^d$ and $\mathbf{E}_{\mathbf{w}_m^i}^{\text{EnCodec}} = \text{Waveform-Encoder}(\mathbf{Z}_{\mathbf{w}_m^i}^{\text{EnCodec}}) \in \mathbb{R}^d$, matching the hidden dimension of the LM. These embeddings are then cross-reconstructed back into their respective representation spaces as $\hat{\mathbf{Z}}_{\mathbf{t}_m^i}^{\text{Mu}} = \text{Symbolic-Decoder}(\mathbf{E}_{\mathbf{w}_m^i}^{\text{EnCodec}})$ and $\hat{\mathbf{Z}}_{\mathbf{w}_m^i}^{\text{EnCodec}} = \text{Waveform-Decoder}(\mathbf{E}_{\mathbf{t}_m^i}^{\text{Mu}})$.

## 4.2 Lanuguage Model

As we define in the problem formulation section, the LM backbone takes multimodal embeddings processed by a unified tokenizer to generate a sequence of textual tokens $a = [a_1, \ldots, a_n]$. The frozen LM parameters are complemented by a LoRA module trained to predict the next output token. It takes the textual embeddings $\mathbf{E_p}, \mathbf{E}_{\mathbf{t}_m}^{\text{LM}}$, and the adapter-wrapped music embeddings $\mathbf{E}_{\mathbf{t}_m^i}^{\text{Mu}}, \mathbf{E}_{\mathbf{w}_m^i}^{\text{EnCodec}}, \mathbf{E}_{\mathbf{w}_m}^{\text{CLAP}}, \mathbf{E}_{\mathbf{w}_m}^{\text{MERT}}$, and its output probability is expressed as:

$$P(\mathbf{a}_i \mid \mathbf{p}, \mathbf{t}_m, \mathbf{w}_m, \mathbf{a}_{1:i-1}) = \text{LM}^{\text{LoRA}}\Big(\mathbf{E_p}, \mathbf{E}_{\mathbf{t}_m}^{\text{LM}}, \text{Adapter}(\mathbf{E}_{\mathbf{t}_m^i}^{\text{Mu}}, \mathbf{E}_{\mathbf{w}_m^i}^{\text{EnCodec}}, \mathbf{E}_{\mathbf{w}_m}^{\text{CLAP}}, \mathbf{E}_{\mathbf{w}_m}^{\text{MERT}}), \mathbf{a}_{1:i-1}\Big) \tag{1}$$

## 4.3 Training Strategy

In order to mitigate the challenge of lacking training data where symbolic music, waveform music, and textual instructions all appear together, we propose a multi-stage training strategy, which includes three consecutive stages: Knowledge Injection (aligning symbolic music and text), Bar-level Alignment (aligning symbolic music and waveform), and MultiModal Fine-tuning (using waveform tasks to align all modalities).

**Stage 1: Knowledge Injection** We begin with using music knowledge and symbolic music datasets to warm up the LM base model. Music encoders are disconnected and symbolic music is merely treated as text. Training is achieved through a negative log-likelihood (NLL) objective, where the model predicts the next token $\mathbf{a}_i$ in the sequence based on the previous tokens $\mathbf{a}_{1:i-1}$:

$$\underset{\Theta_{\text{LoRA}}}{\arg\min} \mathcal{L}_{\text{KI}} = -\frac{1}{l_\mathbf{a}} \sum_{i=1}^{l_\mathbf{a}} \log P(\mathbf{a}_i \mid \mathbf{p}, \mathbf{t}_m, \mathbf{a}_{1:i-1}). \tag{2}$$

**Stage 2: Bar-level Alignment** To align the symbolic and waveform intermediate embeddings within the shared latent space, we apply NCE (Gutmann & Hyvärinen, 2010) as contrastive loss:

$$\mathcal{L}_{\text{NCE}} = -\log \frac{\exp(\cos(\mathbf{E}_{\mathbf{t}_m}^{\text{Mu}}, \mathbf{E}_{\mathbf{w}_m}^{\text{EnCodec}})/\tau)}{\sum_{i=1}^{N} \exp(\cos(\mathbf{E}_{\mathbf{t}_i}^{\text{Mu}}, \mathbf{E}_{\mathbf{w}_i}^{\text{EnCodec}})/\tau)}, \tag{3}$$

where $\cos(\cdot, \cdot)$ represents cosine similarity, $\tau$ is a temperature parameter, and $N$ is the number of negative samples. To mitigate excessive information loss, we apply a cross-reconstruction loss, represented as: $\mathcal{L}_{\text{rec}}^{\text{Mu}} = \|\hat{\mathbf{Z}}_{\mathbf{t}_m}^{\text{Mu}} - \mathbf{Z}_{\mathbf{t}_m}^{\text{Mu}}\|_2^2$, $\mathcal{L}_{\text{rec}}^{\text{EnCodec}} = \|\hat{\mathbf{Z}}_{\mathbf{w}_m}^{\text{EnCodec}} - \mathbf{Z}_{\mathbf{w}_m}^{\text{EnCodec}}\|_2^2$. Thus, the loss for bar-level alignment, which combines both contrastive and reconstruction losses, is denoted as:

$$\underset{\Theta_{\text{Bar}}}{\arg\min} \mathcal{L}_{\text{Bar}} = \mathcal{L}_{\text{contrastive}} + \mathcal{L}_{\text{rec}}^{\text{Mu}} + \mathcal{L}_{\text{rec}}^{\text{EnCodec}}. \tag{4}$$

**Stage 3: MultiModal Fine-tuning** In the final stage, we freeze the bar-level Tokenizer, LoRA-tune the LM and train adapters to accommodate musical representations for all downstream tasks across different datasets that include symbolic music, waveform music, and textual instructions. We formally present the final stage training as follows:

$$\underset{\Theta_{\text{LoRA}}, \Theta_{\text{Adapter}}}{\arg\min} \mathcal{L}_{\text{MFT}} = -\frac{1}{l_\mathbf{a}} \sum_{i=1}^{l_\mathbf{a}} \log P(\mathbf{a}_i \mid \mathbf{p}, \mathbf{t}_m, \mathbf{w}_m, \mathbf{a}_{1:i-1}). \tag{5}$$

## 5 Experiments

In order to evaluate our proposed UniMuLM, we conduct extensive experiments on 9 downstream tasks in terms of multimodal music understanding and generation. We explicate the specific experimental settings and evaluation results as follows.

## 5.1 IMPLEMENTATION

**Hyperparameter Settings.** We employ the Llama3-8B as the LM backbone, with a hidden dimension of 4096, a learning rate of 5e-6, and a total batch size of 16 across 4 devices. We apply a 64-rank LoRA with $\alpha = 16$. For the multimodal tokenizer, the adapter modules consist of a self-attention layer and an MLP, encoding waveform features, $\mathbf{E}_{\mathbf{w}_m}^{\text{CLAP}}$ and $\mathbf{E}_{\mathbf{w}_m}^{\text{MERT}}$, into 8 and 6 tokens, respectively, while each bar feature $\mathbf{E}_{\mathbf{t}_m^i}^{\text{Mu}}$ and $\mathbf{E}_{\mathbf{w}_m^i}^{\text{EnCodec}}$ is encoded as one token, with 4096 dimensions, aligning with the hidden dimension of Llama3.

Table 1: Statistics of training datasets.

| Modality | Dataset | Task | Size | Sampled | Tokens | Waveform Length |
|---|---|---|---|---|---|---|
| Text | MP-Knowledge | KnowledgeQA | 255K | 75K | 63M | - |
| | MP-Summary | Summary | 500K | 5K | 4.2M | - |
| + Symbolic | MP-IrishMAN | Generation | 340K | 60K | 5.4M | - |
| | MP-JSBChorales | Generation | 33K | 30K | 3.2M | - |
| | MP-KernScores | Generation | 10K | 10K | 1.1M | - |
| | MH-continue | Continuation | 820K | 75K | 6.6M | - |
| | MH-inpaint | Inpainting | 820K | 45K | 4.1M | - |
| | MidiCaps | Captioning | 160K | 1K | 42K | - |
| + Waveform | LP-MusicCaps | Captioning | 5.5K | 5K | 280K | 58 Hour |
| | Syn-MidiCaps | Captioning | 160K | 10K | 430K | 280 Hour |
| | SongDescriber | Captioning | 0.7K | 0.5K | 18K | 4.2 Hour |
| | MusicQA | Reasoning | 110K | 10K | 290K | 86 Hour |

**Training Datasets.** Table 1 categorizes the datasets into text, symbolic, and waveform-based. **Size** and **Sampled** denote the total and selected samples for training, **Tokens** is the total token count, and **Waveform Length** represents audio duration in hours. The first category is primarily sourced from MusicPile (MP) (Yuan et al., 2024), with significant cleaning and downsampling applied to its Music Knowledge and Music Summary components to filter out data of low relevance. To address music source bias in MP, we supplemented the symbolic-based datasets with MelodyHub (MH)(Wu et al., 2024), enhancing their diversity. MidiCaps (Melechovský et al., 2024) was converted to ABC format and synthesized into waveforms for both symbolic- and waveform-based captioning (noted as Syn-MidiCaps). Additionally, the waveform-based datasets include LP-MusicCaps (Doh et al., 2023), SongDescriber (Manco et al., 2023), and MusicQA (Liu et al., 2024). Data resample details provided in the Appendix.

## 5.2 QUANTITATIVE EVALUATION

We benchmark UniMuLM across three types of tasks: Music Knowledge Injection, Waveform Music Understanding, and Symbolic Music Generation.

**Music Knowledge Injection.** We evaluate the model's music knowledge using Music-TheoryBench (Yuan et al., 2024), a multiple-choice dataset derived from college-level textbooks and exam materials, as shown in Table 2. To assess the model's ability to handle symbolic music-related questions, we divide the tasks into those with and without Symbolic Notation (SN) and use the 5-majority-vote strategy to ensure more reliable evaluation results.

Table 2: Performance comparison on MusicTheory-Bench (Multiple Choice Question Accuracy).

| Category | Model | w/o-SN | w-SN | Overall |
|---|---|---|---|---|
| General | GPT-3.5 | 0.392 | 0.253 | 0.323 |
| | GPT-4 | 0.567 | 0.308 | 0.437 |
| | GLM4 | 0.539 | 0.285 | 0.402 |
| | Llama2-7B | 0.346 | 0.248 | 0.297 |
| | Llama3-8B | 0.371 | 0.253 | 0.312 |
| MuLM | LTU | 0.363 | 0.243 | 0.317 |
| | ChatMusician | 0.385 | 0.273 | 0.334 |
| UniMuLM | Proposed | **0.613** | **0.393** | **0.503** |
| | w/o Bar-Align. | 0.611 | 0.288 | 0.448 |

Baselines include general LLMs and specialized MuLMs to analyze how parameter scales and training configurations affect music knowledge performance. GLM4 and GPT-4 (Achiam et al., 2023) perform well on tasks without SN, achieving accuracies of 0.539 and 0.567, but show significant declines on SN tasks. Models like LTU (Gong et al., 2024), Llama2 (Touvron et al., 2023), Llama3 (Team, 2024), and GPT-3.5 perform slightly above random chance, reflecting limitations in

Table 3: Performance comparison on music understanding tasks.

| Category | Model | LP-MusicCaps | | SongDescriber | | Syn-MidiCaps | | MusicQA | |
|---|---|---|---|---|---|---|---|---|---|
| | | BLEU | R-L | BLEU | R-L | BLEU | R-L | BLEU | R-L |
| MuLM | LTU | 0.216 | 0.248 | 0.222 | 0.237 | 0.201 | 0.223 | 0.242 | 0.328 |
| | Audio-Flamingo | 0.221 | **0.320** | 0.218 | 0.302 | 0.213 | 0.297 | 0.234 | 0.337 |
| | LLark | 0.278 | 0.250 | 0.243 | 0.237 | 0.248 | 0.268 | 0.201 | 0.194 |
| | Mu-LLaMA | **0.281** | 0.316 | 0.278 | 0.313 | **0.271** | 0.306 | **0.306** | **0.466** |
| UniMuLM | Proposed | 0.262 | 0.302 | **0.281** | **0.334** | 0.260 | **0.308** | 0.285 | 0.401 |
| | w/o Bar-Align. | 0.254 | 0.283 | 0.274 | 0.304 | 0.240 | 0.284 | 0.282 | 0.403 |
| | w/o MERT | 0.207 | 0.248 | 0.273 | 0.318 | 0.241 | 0.261 | 0.244 | 0.339 |
| | w/o CLAP | 0.213 | 0.259 | 0.284 | 0.326 | 0.237 | 0.255 | 0.269 | 0.347 |

music knowledge. While larger models like GPT and GLM leverage extensive world knowledge, ChatMusician, benefiting from sufficient training on SN-related tasks, shows relatively strong performance on w-SN tasks and outperforms open-source models of comparable scale on w/o-SN tasks. UniMuLM, our proposed model, outperforms both general and specialized models, resulting in an overall score of 0.503. Ablation studies highlight the critical role of the bar-level alignment mechanism, which significantly enhances the model's ability to process SN. Without this mechanism, performance on w-SN tasks drops to 0.288, emphasizing its importance for understanding symbolic music features.

**Waveform Music Understanding.** We evaluate the performance of various models on waveform-based music understanding tasks, including LP-MusicCaps (Doh et al., 2023), SongDescriber (Manco et al., 2023), MIDICaps (Melechovský et al., 2024), and MusicQA (Liu et al., 2024), using BLEU (Papineni et al., 2002) and ROUGE-L (R-L) (Lin, 2004) as evaluation metrics, as shown in Table 3. The baseline models (Gong et al., 2024; Kong et al., 2024; Gardner et al., 2023; Liu et al., 2024) all employ adapters to inject waveform features into the large language models. Among the compared models, Mu-LLaMA achieves the highest scores on MusicCaps (BLEU: 0.281, R-L: 0.316) and MusicQA (BLEU: 0.306, R-L: 0.466), demonstrating its strong capability to generate accurate and well-structured descriptions. UniMuLM achieves comparable results, performing weaker on the longer-text LP-MusicCaps but demonstrating better performance on the shorter-text SongDescriber and MusicQA. The ablation study underscores the bar-level alignment module's importance for waveform tasks and the necessity of MERT and CLAP encoding. Removing bar-level alignment (w/o Bar-Align) significantly reduces performance, especially on MusicCaps. Similarly, excluding MERT (w/o MERT) or CLAP (w/o CLAP) degrades performance across all tasks. MusicQA is most impacted by MERT removal, while MusicCapsand SongDescriber are more affected by CLAP removal.

Table 4: Performance comparison on symbolic music generation tasks.

| Category | Model | Continuation | | | | | Inpainting | | | | |
|---|---|---|---|---|---|---|---|---|---|---|---|
| | | Acc | Valid | RC | BLEU | R-L | Acc | Valid | RC | BLEU | R-L |
| General | GPT-3.5 | 0.520 | 0.895 | 0.543 | 0.134 | 0.253 | 0.303 | 0.954 | 0.230 | 0.108 | 0.107 |
| | GPT-4 | 0.586 | 0.912 | 0.645 | 0.341 | 0.556 | 0.330 | 0.963 | 0.255 | 0.122 | 0.282 |
| | GLM4 | 0.520 | 0.865 | 0.602 | 0.301 | 0.471 | 0.342 | 0.910 | 0.243 | 0.123 | 0.273 |
| | Llama2-7B | 0.334 | 0.651 | 0.401 | 0.089 | 0.102 | 0.281 | 0.768 | 0.103 | 0.094 | 0.106 |
| | Llama3-8B | 0.502 | 0.756 | 0.457 | 0.205 | 0.213 | 0.312 | 0.799 | 0.120 | 0.114 | 0.121 |
| MuLM | MuPT | - | 0.694 | 0.197 | 0.120 | 0.172 | - | - | - | - | - |
| | ChatMusician | 0.553 | 0.852 | 0.630 | 0.487 | 0.532 | 0.454 | 0.885 | 0.121 | 0.069 | 0.082 |
| UniMuLM | Proposed | **0.681** | **0.950** | **0.650** | **0.489** | **0.646** | **0.632** | 0.961 | **0.341** | **0.142** | **0.284** |
| | w/o Bar-Align. | 0.652 | 0.948 | 0.638 | 0.482 | 0.598 | 0.612 | **0.965** | 0.321 | 0.132 | 0.265 |

**Symbolic Music Generation.** We evaluate symbolic music generation capabilities on continuation and inpainting tasks, constructed from randomly selected cases in the validation set of the MelodyHub (Wu et al., 2024) dataset. Both tasks include multiple-choice questions, evaluated using 5-majority-vote accuracy (Acc), as well as generation assessed through text-based metrics such as BLEU and ROUGE-L (R-L), and music-specific metrics such as Rhythmic Consistency (RC) and Validity (Valid), as shown in Table 4. RC evaluates rhythm patterns by assigning identical pitches to all notes and calculating BLEU scores, while Validity checks ABC notation syntax, with issues typically involving beat counts or barline errors.

These metrics measure different aspects of the model's music generation ability. Accuracy reflects whether the model can distinguish different musical patterns, Validity assesses if the model understands the format of ABC notation, Rhythmic Consistency evaluates the model's ability to mimic rhythm, while BLEU and R-L directly measure the distance from the ground truth. General models, represented by GPT-4, achieve reasonable performance in continuation tasks, reflecting their strong few-shot learning capabilities with textual input. In contrast, the music-specific MuPT struggles due to the lack of instruction tuning, making it less applicable across multiple tasks. ChatMusician outperforms general models in the continuation task, yet falls short in the more challenging inpainting task, which is absent from its training data, and even struggles to generate rhythmically coherent music compared to general models. UniMuLM excels in generating syntactically valid, rhythmically consistent music, outperforming all baselines. Ablation studies confirm the critical role of the bar-level alignment mechanism, with its absence causing notable declines in RC and accuracy across both tasks.

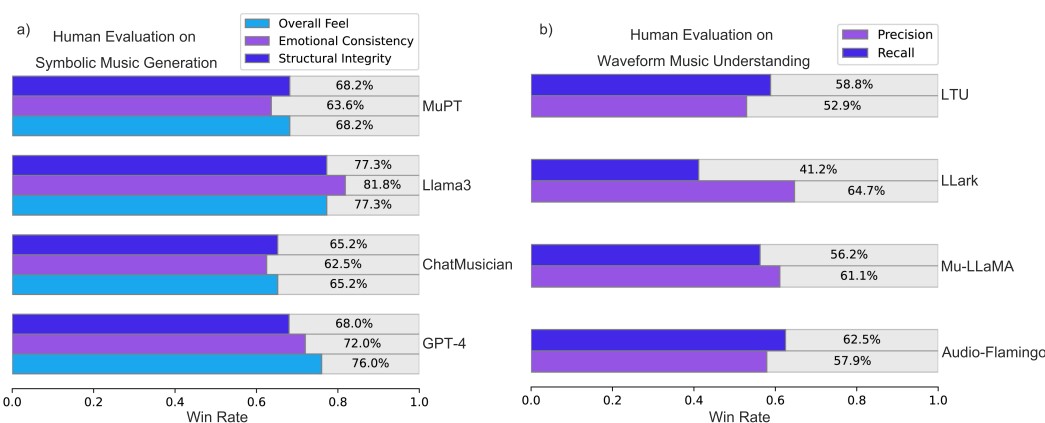

Figure 5: Human evaluation results between UniMuLM and baseline models, where the win rates are calculated from testers' binary ratings on (a) symbolic music generation (assessed by overall feeling, emotional consistency, and structural integrity) and (b) waveform music understanding (based on precision and recall).

## 5.3 HUMAN EVALUATION

Quantitative evaluation has inherent limitations in fully capturing the quality of music understanding and generation. Specifically, for music understanding, while BLEU and ROUGE-L scores offer measurable outcomes, they often cannot distinguish between accurately *capturing key information* and merely *matching a template*. Likewise, in music generation, using ground truth as a reference overlooks the fact that valid, appealing outputs can be diverse and non-unique. To bridge these gaps, we conduct human evaluation.

The results of the human evaluation are shown in Figure 5. For music generation, we randomly select 32 pieces of valid generated music notation for comparison and ask testers to evaluate the generated music in terms of overall feel, emotional consistency, and structural integrity. For the music understanding tasks, we randomly sample data from LP-MusicCaps, SongDescriber, MIDICaps, and MusicQA. For each sample, we provide the results from UniMuLM along with outputs from two of the baseline models. We request that testers rank the three output results based on precision and recall of valid information.

UniMuLM consistently outperforms other models in music generation quality, particularly exceeding the performance of generative models like Llama3 and GPT-4. In terms of music understanding capabilities, UniMuLM's precision surpasses that of existing baselines, while the richness of its output is comparable to other models but falls short of LLark.

## 6 CONCLUSION

In this work, we proposed unifying different music representations (*i.e.,* symbolic and waveform music) and textual instructions with a pre-trained large language model. Specifically, we introduced a novel framework, named UniMuLM, characterized by a unified tokenization process to handle multiple input modalities and specifically model the correspondence between waveform and symbolic representations at the bar level. To train UniMuLM efficiently and effectively, we also applied a multi-stage training strategy to optimize the model on open-source datasets. Extensive empirical results across nine music tasks demonstrate the effectiveness of UniMuLM and underscore the rationale for integrating different music representations. Our work advances the state of MuLMs, where existing works solely rely on a single music representation, to the utilization of multiple representations. Hence, our work paves the way for comprehensive music understanding while contributing to the family of multimodal language models.

## 7 DISCUSSION

**Limitations.** Despite UniMuLM achieves remarkable performance across various music tasks, there are three main limitations: (1) symbolic infomation lossing (2) limitations in the robustness of bar-level alignment module and (3) constrained training scale. The first limitation arises from the simplification of key signatures in symbolic music training. To ensure training stability, our model converts all major keys to C and minor keys to A minor to match the actual pitch during performance. However, this risks losing key-specific features that are critical for capturing tonal subtleties and preserving the emotional and structural identity of the music. Such an approach can obscure the distinct characteristics and significance of different musical keys. The second limitation lies in the construction of the bar-level alignment module. Currently, it only processes single-track music synthesized with a single instrument. While effective, this setup lacks the ability to handle multi-track compositions or diverse instrumentation, and it does not account for real-world scenarios involving noisy, non-synthetic music. These factors limit the robustness and generalizability of the model. The third limitation is related to the model's quantization and fine-tuning methods. To ensure experimental efficiency, we applied 4-bit quantization and used LoRA for fine-tuning, which may constrain the model's performance. Although we trained the model on approximately 80 million tokens—surpassing typical LoRA tuning scenarios—full-scale supervised fine-tuning may be more suitable in some cases. Additionally, only a portion of the available dataset was used, which limits the model's potential. Leveraging larger models and more extensive datasets could significantly enhance music understanding, offering opportunities for improved performance through broader data usage and alternative fine-tuning methods.

**Future Work.** Future work will focus on four directions: (1) retaining key-specific features, (2) enhancing bar-level alignment with multi-track and real-world music, (3) scaling up the model and dataset, and (4) generating waveform music end-to-end. First, we will encode both key-biased and key-unbiased notation representations, thus enhancing the current approach by adding explicit key feature extraction. Second, we will improve the bar-level alignment module by incorporating multi-track compositions and using multiple synthesizers to generate training data. Furthermore, future iterations will include real-world, noisy music to enhance the model's robustness. This approach will ensure that the alignment mechanism better reflects the complexity and diversity of actual musical scenarios, making the model more versatile and reliable. Third, we will explore using higher-bit quantization or full-scale SFT to improve model performance. Since the quality and scale of SFT data significantly influence the model's effectiveness, efforts should be made to develop fine-grained datasets that better capture temporal structures in music. At the end, we will extend the existing model to generate waveform music end-to-end, thereby broadening the application of UniMuLM. For example, it could support a wider variety of symbolic music notations, such as MIDI, or integrate codified waveform music, such as EnCodec tokens, as interleaved inputs, enabling the model to directly generate waveform music.

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

## A    ETHICS STATEMENT

### A.1    USE OF OPEN-SOURCE DATASETS

The training datasets used in this study are categorized into pure text, symbolic-based, and waveform-based. All datasets were sourced from publicly available repositories: MusicPile[1], MusicTheoryBench[2], MelodyHub[3], MidiCaps[4], LP-MusicCaps[5], SongDescriber[6], and MusicQA[7]. All data preprocessing steps ensured compliance with licensing terms, and no proprietary datasets were used in this study.

### A.2    POTENTIAL IMPACTS OF GENERATED CONTENT

The model's ability to generate music introduces several potential risks that warrant careful consideration:

**Bias and Cultural Sensitivity**: Generated content may inadvertently reflect biases present in the training data, as the datasets used could inherently carry cultural, stylistic, or demographic imbalances. These biases might result in music that favors certain genres, styles, or cultural norms while neglecting others. Additionally, the model might struggle to fully capture the nuances and subtleties of diverse musical cultures, potentially leading to outputs that are perceived as stereotypical, insensitive, or unrepresentative.

**Misuse and Ethical Concerns**: The model's capabilities could be misused to produce music that is culturally inappropriate, offensive, or plagiarized. The model could produce content intended to mock or degrade specific cultures or communities, exacerbating ethical concerns around the responsible use of AI in creative domains.

**Copyright Issues**: Although the datasets used in this research are open-source and comply with licensing terms, there remains a risk of the model generating outputs that inadvertently resemble

---

[1]https://huggingface.co/datasets/m-a-p/MusicPile
[2]https://huggingface.co/datasets/m-a-p/MusicTheoryBench
[3]https://huggingface.co/datasets/sander-wood/melodyhub
[4]https://github.com/AMAAI-Lab/MidiCaps
[5]https://github.com/seungheondoh/lp-music-caps.git
[6]https://huggingface.co/datasets/renumics/song-describer-dataset
[7]https://huggingface.co/datasets/mu-llama/MusicQA/tree/main

copyrighted works. This could occur due to overfitting on specific pieces of training data or the model's reliance on patterns present in the source material.

## B    IMPLEMENTATION OF BASELINES

**General-Purpose Models** We include several widely used large language models as baselines for single track symbolic music generation:

- **GPT-3.5**[8]: Known for its strong language understanding and reasoning capabilities, GPT-3.5 serves as a robust baseline for tasks requiring textual and symbolic comprehension.

- **GPT-4**[9]: As an advanced version of GPT-3.5, GPT-4 incorporates improved reasoning and multimodal capabilities, making it a competitive model for text-heavy music tasks.

- **Llama2-7B**[10]: This open-source model is recognized for its efficiency and effectiveness in general language understanding, making it a solid choice for evaluating text-based music tasks.

- **Llama3-8B**[11]: As a scaled-up version of Llama2, this model provides additional capacity for handling complex reasoning tasks, serving as a strong baseline.

- **GLM4**[12]: A versatile general-purpose model optimized for multimodal tasks, GLM4 bridges textual and contextual understanding, enabling comparisons on multimodal music tasks.

**Music-Specific Models** We compare UniMuLM with several models designed for symbolic or waveform-based music understanding and generation:

- **ChatMusician**[13]: This model specializes in symbolic music generation and understanding, leveraging MusicPile[14] datasets for training.

- **MuPT**[15]: A purely symbolic music model based on a decoder-only Transformer architecture, trained from scratch. MuPT excels in melody generation and continuation tasks but lacks natural language understanding.

- **LTU**[16]: Trained on a 5M audio QA dataset, LTU exhibits general understanding and reasoning capabilities for both audio and music.

- **Audio-Flamingo**[17]: Incorporats xattn-dense layers from Flamingo[18] to condition on audio inputs effectively.

- **LLark**[19]: Trained on language model-enhanced music metadata, utilizing CLAP as an encoder, and achieving impressive results in tasks such as key estimation, tempo estimation, genre classification, and instrument identification.

- **Mu-LLaMA**[20]: Trained on the MusicQA dataset of open-ended music-related questions. It integrates MERT features into a LLaMA backbone via an adapter and excels in music-related QA tasks.

**Other Baselines** We have prioritized implementing the above baselines. However, there are several notable works that are worth attention and may be included in future revisions:

---

[8] https://platform.openai.com/docs/models#gpt-3-5
[9] https://platform.openai.com/docs/models#gpt-4
[10] https://huggingface.co/meta-llama/Llama-2-7b
[11] https://github.com/facebookresearch/llama
[12] https://huggingface.co/THUDM/glm-4-9b-chat
[13] https://github.com/m-a-p/ChatMusician
[14] https://huggingface.co/datasets/m-a-p/MusicPile
[15] https://huggingface.co/m-a-p/MuPT-v1-8192-1.97B
[16] https://github.com/YuanGongND/ltu
[17] https://github.com/NVIDIA/audio-flamingo.git
[18] https://github.com/deepmind/Flamingo
[19] https://github.com/spotify-research/llark.git
[20] https://github.com/shansongliu/MU-LLaMA.git

- **GPT-4o**[21]: An advanced extension of GPT-4 designed for optimized performance in multimodal reasoning tasks.

- **SALMONN**[22]: A model with LoRA tuning from Vicuna LLM, designed for general audio tasks including speech, audio events, and music. It employs a window-level Q-Former as the adapter. The authors claim that, with relatively low training overhead, it retains and regains the emergent abilities of the original model.

- **Qwen-Audio2**[23]: A Large-Scale Audio-Language model for general audio tasks, with a focus on enabling multi-turn dialogues and supporting diverse audio-oriented scenarios.

- **MusiLingo**[24]: Employ MERT-330M as the music encoder and Vicuna-7B as the language model. Trained on created MusicInstruct datasets which features 60,493 Q&A pairs covering both general questions like music summarisation, and specific questions related to music genres, moods, and instruments.

## C   HUMAN EVALUATION DETAILS

For the music generation evaluation, we select 32 music pieces generated by UniMuLM and four baseline models (GPT-4, Llama3, MuPT, and ChatMusician), synthesized using Piano, Flute, Saxophone, and Violin to capture a range of melodic and timbral features. Testers were asked to assess each piece across three categories: the overall aesthetic quality or feel of the music, the emotional consistency conveyed throughout the piece, and the structural integrity in terms of coherence and logical progression.

For the evaluation of music understanding, we sampled data from multiple benchmark datasets, including LP-MusicCaps, SongDescriber, MIDICaps, and MusicQA. For each task, the outputs generated by UniMuLM were presented alongside the results from two randomly selected baseline models. Testers were instructed to rank the models' outputs based on two key criteria: precision, which measures the relevance of the information to the given query, and recall, which evaluates the completeness of the meaningful content captured. This process aimed to assess the extent to which each model effectively captured critical details while minimizing irrelevant or extraneous information.

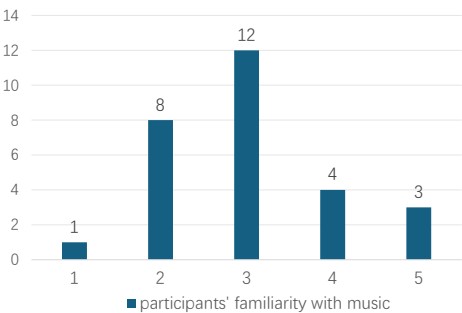

Figure 6: Distribution of participants' familiarity with music. Scores range from 1 (minimal exposure) to 5 (formal training).

We collect responses from 28 participants. To assess their familiarity with music, we included a preliminary question: "How much do you engage with music?" The scoring scale ranged from 1 to 5, where 1 indicated minimal exposure (e.g., rarely listening to music), 2 represented frequent music listening, 3 implied the ability to play an instrument or participation in activities like a choir, 4 denoted proficiency in at least one musical instrument, and 5 indicated formal training in music theory or composition. The distribution of participants' scores is visualized in Figure 6. This distribution aligns with or slightly exceeds the general population's level of musical appreciation. Based on this,

---

[21] https://platform.openai.com/docs/models#gpt-4o
[22] https://github.com/bytedance/SALMONN.git
[23] https://github.com/QwenLM/Qwen2-Audio.git
[24] https://github.com/zihaod/MusiLingo

we opted not to apply weighted adjustments to their ratings and treated all participants' responses with equal weight.

