# OpenReview forum: "Unified Music-Language Model for Symbolic and Waveform Integration"
_ICLR.cc/2025/Conference — Submitted to ICLR 2025_

### Official Review · Reviewer_N56W · 2024-11-01

**Soundness:** 2
**Presentation:** 2
**Contribution:** 3
**Rating:** 3
**Confidence:** 4

**Summary:**

The paper presents UniMuLM, a unified music language model designed to address the limitations of existing music language models (MuLMs) that typically rely on a single representation type. UniMuLM integrates symbolic music, waveform audio, and textual instructions through a novel bar-level tokenizer to facilitate fine-grained, multi-modal music understanding. To handle the complexities of multimodal alignment and stabilize training across these varied representations, the authors implement a multi-stage training strategy. Their empirical evaluation on nine music-related tasks indicates UniMuLM’s performance improvements over current state-of-the-art methods, underscoring the benefits of unified, multi-representational learning for advancing MuLMs.

**Strengths:**

The authors introduce a novel and well-motivated approach that leverages multiple music representations for a unified model, which is an important advancement for multimodal music understanding. The proposed method and experimental design appear to be robust, addressing significant challenges within the field and offering promising results on music theory, acoustic music understanding and symbolic music generation tasks.

**Weaknesses:**

The paper’s primary claims highly overstate the experimental outcomes due to the following reasons.
* The claim of 9 music tasks might be miscounting. Or maybe the author (over)claims the four music captioning datasets as four tasks (theory, caption, QA and generation). According to my understanding, it should be a music theory benchmark, three music caption/description datasets, 1 musical, and 2 types of music generation with 2 different types of evaluation methods. Please clarify what are the nine different tasks.
* as the evaluation lacks comparisons with recently released in the past 9 months, advanced baselines, such as SALMONN, GPT-4o and Qwen2-audio, are relevant to benchmark improvements, which may provide much better results on music theory and music captioning. For example, Qwen-audio and SALMONN tech reports include the sota performance on music captioning and GPT-40 is well-known for its audio instruction following capability
* Additionally, several tasks are missing comprehensive evaluation metrics (e.g., BERT-score and METEOR for music captioning, which are widely used and much more persuasive compared to the BELU score reported in this paper).
* The paper claims the alignment of 3 modalities, but it does not explore the direct alignment of symbolic and audio modalities without intermediate text, e.g. audio transcription to ABC notation, which limits insights into tasks.
* Further, ablation studies are absent for the loss functions introduced in stage two of training, leaving uncertainty around the necessity and optimal weighting of each component. This does not make the methodology proposed in stage 2 solid. You can run the experiments on changing the loss weights or delete part of the loss
* The author claims the impact of bar-level tokenization. However, there is no ablation study on not using such tokenization. Besides, the author does not clarify which dataset requires bar-level information for the model to evaluate. Please clarify why the 4 or 9 tasks are contributed by the bar-level information you provided and show the experimental results if the bar-level tokens indeed help. Maybe it does not align well or screw up the performance by increasing the length of tokens

**Questions:**

Typographic and Formatting Issues:
* The numbering in the paper is inconsistent (e.g., Section 3 contains only one paragraph, Section 4.1 includes only 4.1.1).
* Figure 1’s color distinctions are difficult to discern, and Figure 3(a) is complex, potentially making it harder to interpret than the accompanying text.
* Formulae could benefit from simplification. Besides, some notation feels excessive or potentially confusing. For instance, is “Zbar” intended to be “Z_{bar}”?
* In Table 1, could the authors clarify the meanings of "Quantity" and "Sampled"? Do these refer to token count and instruction-answer pairs?
* Many typo errors

Experimental & Dataset Details:
* Please clarify the precision during training. Is the 4-bit training precision sufficient? Half-precision training is typically float16 (16-bit), and 4-bit precision may impact training stability or gradient calculations. Did the authors consider gradient overflow or stability issues, or was 4-bit precision specifically chosen for other reasons? (Or maybe 4B=32bit, which is the common training precision)
* How was MIDI data transformed into ABC notation, and how was bar/beat-level annotation achieved? Transcribing MIDI to bar-level music sheet is not trivial. The quality of this annotation could significantly impact bar-level information integrity.

**Details Of Ethics Concerns:**

* The study lacks mention of ethical review or participant demographics, particularly if invited participants with diverse music backgrounds were included.
* The paper utilizes several music datasets without clarifying their copyright statuses. For instance, some datasets, like LP-MusicapsMSD, are not publicly accessible, raising questions about how the authors secured permissions or accessed copyrighted content for this research.

---

> ### Author Response · Authors · 2024-11-29
>
> Thank you for your numerous suggestions.
>
> **W1**: After careful reconsideration, we acknowledge that our work encompasses five tasks across nine test datasets. These tasks include music theory, waveform-based tasks (captioning and QA), and symbolic tasks (continuation and inpainting). While "generation" may not fully describe certain tasks—such as inpainting, which is typically not classified under generation—our current focus remains on single-track melodies. To better reflect these distinctions, we will adjust the terminology in future revisions.
>
> In fact, from our perspective, we do not intend to overly highlight the generation tasks. In our experiments, continuation and inpainting primarily serve to maximize the utilization of unlabeled symbolic music data. In other words, these are pseudo-tasks designed for training purposes. Our model primarily emphasizes music theory and understanding capabilities. Both symbolic music generation and waveform music generation, as mentioned in the future work section, are considered downstream applications of this work. We will make this distinction more explicit in future versions.
>
> **W2**: Our current baseline selection primarily considers structural and dataset similarities to our model. We have not yet included GPT-4o, Qwen-Audio2, and SALMONN as baselines because their model scales and training data sizes differ significantly from ours. Also, they have not reported results on music understanding benchmarks, are instead focused on traditional MIR tasks.  Consequently, we did not prioritize reproducing their results for this version. However, we will ensure their inclusion in future comparisons to provide a more comprehensive evaluation.
>
> **W3**: We have evaluated the tasks using additional metrics, including BERT-score, METEOR, and CIDEr. These results will be included in the Appendix of the revised version.
>
> **W4**: We conducted preliminary experiments on transcription tasks but did not include them in the results for two reasons. First, we initially considered transcription as atypical for MuLMs. Second, specialized models optimized for this task naturally outperform general-purpose approaches like ours. However, as you pointed out, transcription is indeed valuable for demonstrating alignment between waveform and symbolic modalities. We will include this task in future experiments to further validate our model's alignment capabilities.
>
> **W5**: Thank you for your guidance. In stage 2, training a reliable low-level encoder that minimizes information loss is indeed non-trivial. We relied on empirical observations and experimental results without conducting systematic experiments to support this. Due to resource constraints, we did not progress stage 3 training for some stage 2 results that showed poor performance. We will address this gap in future iterations and provide more rigorous evaluations.
>
> **W6**: While we do not have a separate ablation study section, we have reported results for "w/o bar-align" in the experimental results tables for each task. However, as you rightly noted, we failed to provide a detailed explanation of its impact. To address this, we will include additional case studies in future versions to better illustrate how bar-level tokenization contributes to specific tasks. However, as you pointed out, transcription is indeed a valuable task for demonstrating the alignment between waveform and symbolic modalities. We will include this task in future experiments to further validate our model's alignment capabilities.

---

> > ### Author Response · Authors · 2024-11-29
> >
> > **Q1**: Thank you for your valuable suggestions regarding the representation in our paper. We will revise it according to your feedback.
> >
> > **Q2.1**: 4-bit precision was specifically chosen to accommodate the large size of our model and dataset within available computational resources to efficient training. Empirical results demonstrated that this approach maintained stability and achieved competitive performance compared to higher-precision training. However, we acknowledge that this method may not be optimal for all tasks or setups and are open to further suggestions on this matter.
> >
> > **Q2.2**: MIDI to ABC conversion is indeed non-trivial. We used (github.com/marmooo/midi2abc) to convert MIDI files into multi-track ABC format, retaining only the melody track for our tasks. During this process, we discarded samples where the conversion failed or the melody track was not prominent to ensure data quality. Additionally, we synthesized MIDI into waveform to create the corresponding waveform dataset. This dual usage allowed us to leverage both symbolic and waveform representations while maintaining alignment across modalities. We will include more details about this process in the Appendix of the revised version.
> >
> > **Ethics Concerns**:
> >
> > For participant, we surveyed 28 participants, asking, "How much do you engage with music?" on a scale of 1 to 5: 1 (1') minimal exposure; 2 (8') frequent listening; 3 (12') playing an instrument or participating in activities like a choir; 4 (4') proficiency in one or more instruments; 5 (3') formal training in music theory or composition. This distribution aligns with or exceeds general population levels, so we treated all responses equally without weighting.
> >
> > For Ethics Concerns, we will include an ethical review statement in the revised version. Regarding dataset accessibility, the LP-MusicCaps dataset is publicly available (\url{https://github.com/seungheondoh/lp-music-caps}), and we ensured compliance with its usage terms. We will also provide detailed information on the accessibility and licensing of all datasets used in our research in Appendix.
> >
> > Thank you again for your valuable feedback.

---

> > > ### Comment · Reviewer_N56W · 2024-11-29
> > >
> > > Thank you for the explanation. I do not see the update in the appendix. Do you upload a new version of the PDF by 27th Nov AOE? If so, I will contact the meta-reviewer for the updated information.
> > >
> > > Please let me know once you have more experiment results.
> > >
> > > How do you implement the 4-bit training? How do you evaluate its stability of training and model capability compared with 8/16/32-bit training?

---

### Official Review · Reviewer_RpSP · 2024-11-01

**Soundness:** 2
**Presentation:** 1
**Contribution:** 2
**Rating:** 3
**Confidence:** 3

**Summary:**

The paper presents a fine-tuned language model for music understanding and generation. The music representations used in this paper are text-like ABC format, audio format, and a representation jointly learned from symbolic and audio domain music data. The input to the model is text and music, and the output is text or music in ABC format. The model demonstrates superior performance in some tasks in generation and understanding.

**Strengths:**

The proposed model incorporates data from multiple modalities and provides a general interface for generation and understanding. Music features from the audio domain and the symbolic domain are considered and proved helpful for the task.

**Weaknesses:**

1. The title is misleading, as it suggests the language model itself integrates multiple modalities, whereas the actual integration occurs primarily at the tokenization level.
2. The paper's writing lacks clarity and rigor. Figure 3 is confusing because there are no distinctions between inputs and outputs and no explanation of color coding for adapters and tokens. The notation is sloppy; symbols like L_c appear in the figure but are not defined in anywhere else in the paper. In the main text, terms like LM(), Adapter(), and Symbolic-Encoder() are presented as informal expressions rather than proper mathematical functions.
3. The integration of audio and symbolic data is bounded by the fact that paired audios are synthesized. The quality in the demo page is not convincing.
4. Figure 1 feels promotional, yet it's hard to tell what are the nine tasks after finish reading (e.g., what is MCQ-inpainting). Additionally, the experiments only assess performance at a superficial level. The paper would benefit from deeper analysis to demonstrate what the model actually gains from modality integration. Providing additional demos, case studies, and comparisons either in the appendices or on the demo page would strengthen the evaluation.

**Questions:**

Questions are mentioned in the weaknesses.

---

> ### Author Response · Authors · 2024-11-29
>
> Thank you for your suggestions.
>
> **W1**: We would like to clarify that in our title, "Unified" is intended to highlight the integration of symbolic music and waveform modalities, rather than a comprehensive unification of music and language. We understand that this phrasing may have caused some misunderstanding, and we will revise it in future iterations. Regarding our problem formulation, since the outputs for ABC notation and music understanding tasks are textual, we believe our definition is accurate and not overstated; however, we will further refine our descriptions to reduce any potential misinterpretations.
>
> The main technical innovation of this work is indeed at the tokenizer level. However, the tokenizer serves as a foundation for the subsequent integration of natural language and music modalities during training. We believe that a key challenge in multimodal language models lies in aligning modalities, resolving ambiguities, and addressing the sparsity or absence of certain modalities—areas where we aim to contribute meaningfully. We would also like to emphasize the role of our bar-level tokenizer, which enables different modalities to benefit from each other’s tasks and datasets during training. This, we believe, is where integration truly happens. Of course, we warmly welcome further discussion with you on this topic.
>
>
> **W2**: Thank you for your suggestions regarding the details of our presentation. We will update the framework diagram and provide formal definitions for LM(), Adapter(), and Symbolic-Encoder() to enhance clarity in future revision.
>
> **W3**: We agree that the integration is currently bounded by the fact that paired audios are synthesized, which may impact robustness. However, given our resource constraints, this is a necessary trade-off. Synthesized audio, with its high accuracy and consistency, remains a reasonable choice for alignment training, as demonstrated by the effectiveness of our approach in experiments. We will enhance the demo page by including more comparisons with other baselines to provide a more intuitive demonstration of the model's advantages. And, we will consider enhancing robustness, as mentioned in our response to Reviewer 2's Weakness 3.
>
> **W4**: We acknowledge that Figure 1 could be clearer in conveying the five types of tasks and nine test datasets. We will revise its caption to include more explicit explanations. Regarding the experiments, since the improvements stem from additional modality integration and training data, conducting detailed ablation studies may be challenging. We will enhance the demo page and include tasks like music transcription and traditional MIR tasks (e.g., key estimation, tempo estimation) to provide more intuitive evaluations of fine-grained alignment and music understanding capabilities.
>
> Thank you again. We are committed to improving our work based on your suggestions.

---

### Official Review · Reviewer_dR3n · 2024-11-03

**Soundness:** 3
**Presentation:** 3
**Contribution:** 2
**Rating:** 6
**Confidence:** 4

**Summary:**

This paper introduces a Unified Music-Language Model that integrates symbolic music and waveforms through bar-level tokenization inspired by bar-patching [1], addressing the issue that current music language models struggle to handle both notation and performance due to temporal scale inconsistencies.
The proposed approach follows a three-stage training strategy. First, it incorporates music knowledge to warm up Llama-3 with foundational music theory concepts. Second, a bar-level tokenizer is trained on paired symbolic and waveform data using contrastive and reconstruction losses. Finally, LoRA-tuning is applied to adapt the model for various downstream tasks involving diverse music representations.
By uniting these two modalities, the model enhances both music understanding and generation capabilities, showing advantages over single-modality models in three downstream tasks: music theory injection, waveform-based music understanding, and symbolic music generation.

[1] Shangda Wu, Dingyao Yu, Xu Tan, and Maosong Sun. Clamp: Contrastive language-music pre-training for cross-modal symbolic music information retrieval. In ISMIR, pp. 157–165, 2023.

**Strengths:**

This paper introduces a music-language model capable of encoding cross-modality data for symbolic music and waveforms. It addresses the challenge of temporal consistency by using a bar-level tokenizer to align music waveforms with notation, employing contrastive and reconstruction losses to enhance alignment between symbolic and waveform information.

The paper is well-written, with a clear and well-defined problem statement that makes the methodology and contributions straightforward and easy to understand. This work offers a valuable approach to unifying multiple input modalities in music through a unified tokenizer, paving the way for enhanced music understanding and more controllable music generation.

**Weaknesses:**

1. Limited Novelty in Modality Alignment: This paper is not the first to align audio waveforms with symbolic representations. For example, JASCO [1] employs ‘nearest’ interpolation for chords and ‘linear’ interpolation for melody, resampling them to match EnCodec’s frame rate. To strengthen the paper’s contribution, it would be helpful to emphasize the specific advantages offered by your alignment strategy. For example, how your bar-level tokenization differs from or improves upon interpolation-based approaches in terms of preserving musical structure or handling different types of musical elements.
[1] Or Tal, Alon Ziv, Itai Gat, Felix Kreuk, and Yossi Adi. “Joint Audio and Symbolic Conditioning for Temporally Controlled Text-to-Music Generation.” arXiv preprint arXiv:2406.10970, 2024.

2. Marginal Improvement on Waveform Music Understanding Tasks: The model demonstrates limited improvement over Mu-LLaMA on 3 out of 4 datasets for waveform music understanding tasks. This raises questions about the actual benefit of incorporating symbolic information to enhance waveform audio understanding. Providing further exploration or justification of the advantages of symbolic data for audio understanding would strengthen the paper. For example, you can provide a more detailed analysis of where and why your model shows improvements or limitations compared to Mu-LLaMA  and discuss specific examples or task types where symbolic information seems to help or hinder performance.

3. Ignorance of Difference between Real-world Waveform and Synthesized Waveform: the alignment stage does not train on the real-world waveform, which might perform different from the synthesized waveform. I understand large-scale pair data lacks, but you can still use some data augmentation strategies such as using different soundbanks to render the symbolic music, or applying some transcription tools (e.g. MT3[2]) to get the coarse symbolic representation and fine-grain them to ensure valid format via GPT-4. I think it would be better to discuss in your paper the potential impact of using synthesized vs. real-world waveforms on their model's performance.
[2] Gardner, J., Simon, I., Manilow, E., Hawthorne, C., & Engel, J. (2021). MT3: Multi-task multitrack music transcription. arXiv preprint arXiv:2111.03017.

4. Minor Typos: There are minor typos in the description of Figure 3, such as “constrastive loss” and “corss-reconstruction losse.” In Section 4.3, symbols in formula (4) do not correspond with the symbols in the textual explanations above.

**Questions:**

1.	Dataset Construction Process: Could you clarify the dataset construction process, particularly regarding any differences in cleaning and downsampling strategies across datasets? A detailed explanation of the sampling methods used for each dataset would enhance transparency.

2.	Multi-Track Music Processing Details: While the paper discusses bar-level tokenization for single-track music, it lacks details on aligning multi-track waveforms with ABC notation. Could you provide more information on how alignment is managed for multi-track music?

3.	Exploration of General Model’s Music Knowledge: Given that models like GPT-4 and GLM-4 are noted for their in-context learning capabilities, have you explored prompt engineering with these models to assess their music theory knowledge? This could offer a fairer basis for comparison.

4.	Music Understanding Performance Analysis: The UniMuLM model performs better on shorter-text datasets compared to longer-text ones. Could you elaborate on this performance variation? Additional analysis would provide valuable insights into the underlying factors.

5.	More Details about the Subjective Evaluation: how many participants joined the subjective test? What are their music backgrounds? What’s more, your demo page only provides single-track samples and what about the multitrack generation results? How do you ensure consistency across different raters.

6.	About the Waveform Music Generation: intuitively, when seeing your paper’s title, I expect to see the comparison between text-waveform and text-symbolic generation. From my perspectives, that’s the main challenge and concern of symbolic-waveform music alignment work, to support the controllable waveform generation and more diverse symbolic generation simultaneously by getting the embeddings from two domains closer. Therefore, if possible, please discuss what are your initial findings, results or the challenges you've encountered within the text-waveform task, as mentioned in your future work part. It really benefits the whole AI music community.

---

> ### Author Response · Authors · 2024-11-29
>
> Thank you for your numerous suggestions and for acknowledging our idea of bar-level tokenization.
>
> **W1**: We have been aware of JASCO, and we note the following key differences: 1. Unlike JASCO's joint conditioning approach, we align waveform and symbolics as pre-aligned embeddings. 2. Our approach directly uses raw musical notes, avoiding the resampling and interpolation required in JASCO, making it more faithful to the original structure. We will consider adding it to the literature review. And, we will include transcription experiments to showcase our ability to preserve musical structure.
>
> **W2**: We believe our model's suboptimal performance stems mainly from two factors: training resource constraints—we used LoRA instead of full fine-tuning, had less data, and trimmed answer lengths—and existing benchmarks often encourage template fitting over genuine understanding. To address this, we plan to analyze this issue in detail and introduce traditional MIR tasks (e.g., key estimation, tempo estimation, genre classification, instrument identification) to evaluate MuLMs' music understanding capabilities.
>
> **W3**: We agree that relying solely on synthesized waveforms may limit our model's applicability to real-world audio. Your generous suggestions have pointed us in the right direction, and we will dedicate time and resources to implement them. Strategies like rendering symbolic music with diverse soundbanks or using transcription tools are promising avenues for future exploration.
>
> **Q1**: We will provide a detailed explanation of the dataset construction process in the revised version. Briefly, for MusicQA, we excluded questions unrelated to music. For MusicPile's knowledge-related questions, many were only weakly related to music (e.g., concert etiquette). Similarly, MusicPile-Summary mainly focuses on metadata and lyrics summarization, which we believe has limited relevance to music understanding. We retained a small subset to preserve the model's general language abilities. For the other datasets, we aimed to reduce computational costs under limited resources and balance the proportion of tasks in the training set. These decisions optimized training efficiency while ensuring diverse task coverage.
>
> **Q2**: Our current tasks focus on single-track music, i.e., melodies, and do not handle multi-track alignment. We believe melody is more crucial for understanding tasks, and multi-track alignment may be challenging, but we will endeavor to explore it in future work.
>
> **Q3**: Yes, we considered prompt engineering to assess general models' music theory knowledge but found it challenging to design fair comparisons due to the uniqueness of each question in our dataset. We experimented with in-context learning using 5-shot examples, but improvements were limited or even negative. We believe that music inpainting and continuation inherently test ICL capabilities, and that incorrect examples can be detrimental.
>
> **Q4**: We will provide more detailed numerical analysis in revision. Briefly, due to limited training resources, the dataset responses are truncated, and UniMuLM tends to output shorter answers.
>
> **Q5**: We surveyed 28 participants, asking, "How much do you engage with music?" on a scale of 1 to 5: 1 (1') minimal exposure; 2 (8') frequent listening; 3 (12') playing an instrument or participating in activities like a choir; 4 (4') proficiency in one or more instruments; 5 (3') formal training in music theory or composition. This distribution aligns with or exceeds general population levels, so we treated all responses equally without weighting. For multitrack generation, while the model is not currently trained for it, we are working on prompt engineering to include multitrack results in the demo and better showcase its potential.
>
> **Q6**: We acknowledge that the current title may be somewhat misleading. We agree that the complementarity between symbolic and waveform generation is valuable to pursue, but we recognize that the alignment among the two and language is still weak. Therefore, we aim to first focus on music understanding.
>
> Supporting both symbolic and waveform generation is indeed an important goal. When decoding symbolic and text data simultaneously, two primary approaches are feasible: interleaving music tokens, as exemplified by AnyGPT, or leveraging tools for downstream music generation, as seen in NExTGPT. The former often struggles under limited computational resources, while our current efforts focus on the latter approach, using fine-grained control of MusicGen.
>
> Thank you again for your valuable feedback. We are committed to improving our work based on your suggestions.

---

### Official Review · Reviewer_iF7b · 2024-11-04

**Soundness:** 3
**Presentation:** 2
**Contribution:** 2
**Rating:** 5
**Confidence:** 4

**Summary:**

The paper presents UniMuLM, a Unified Music-Language Model designed to integrate symbolic music, waveform music, and textual instructions into a cohesive framework. The model addresses the challenges posed by the distinct representations of music (symbolic notation vs. audio waveforms), aiming to leverage their complementary strengths. Key contributions include:

Unified Tokenization: Introduction of a bar-level tokenizer that aligns symbolic and waveform music representations to enable fine-grained, mutually reinforced understanding.

Multi-Stage Training Strategy: The model is trained in three stages to inject knowledge, align music representations, and fine-tune the system for various music-related tasks.

Comprehensive Evaluation: Demonstrates state-of-the-art (SOTA) performance across different music tasks, validating the effectiveness of integrating multiple music modalities.

**Strengths:**

Comprehensive Multimodal Integration: The model successfully combines symbolic and waveform music representations, addressing a major limitation in existing models and enabling better music understanding and generation.

Innovative Bar-Level Alignment: The bar-level tokenizer offers a novel approach to synchronizing symbolic and audio representations, improving the model’s ability to process and generate music in a contextually relevant manner.

**Weaknesses:**

1. Inconsistency and Misalignment in Title, Problem Formulation, and Paper Focus

The title, "Unified Music Language Model," suggests a comprehensive system capable of generating audio, which is misleading since the model does not generate audio directly. Instead, it is an "audio-informed, text-based music language model" that can handle both music description and symbolic music generation.
The problem formulation further contributes to the confusion, as it mainly addresses a music description problem. However, the actual contributions focus more on using audio information to enhance symbolic music generation and music understanding, indicating a disconnect between the proposed problem, the title, and the work's real impact.


2. Suboptimal Baselines and Limited Impact of SOTA Claims

The choice of baselines for music generation, such as ChatMusician and MUPT, undermines the significance of the model's claimed state-of-the-art performance. Both baselines are first-of-its-kind general-purpose multimodal music models, but with subpar generation quality compared to dedicated symbolic generation models like Music Transformer or the more advanced whole-song generation via hierarchical diffusion models.

A similar issue exists in the music understanding benchmarks. Using Mu-LLaMa as a baseline, while suitable for demonstrating language model integration, fails to compare favorably against specialized Music Information Retrieval (MIR) tools, which excel in task-specific performance. The broader question remains whether integrating music information into a text-based language model leads to genuinely superior performance.

Ultimately, the novelty of integrating music data into language models has become less groundbreaking. The field has matured, and the critical evaluation should focus on whether this integration yields better performance. Based on the provided demos, the symbolic music generation quality lags behind specialized models, and in music QA tasks, errors were evident, as seen in 2nd and 3rd showcased examples.

**Questions:**

1. one main weakness of existing text-based music LM (e.g. chatmusician) is that they can only answer the type of questions that they have been trained on (either in the pertaining stage or instruction fine-tuning state) and fail to generalize on new types of questions. Have you looked into this problem? (I am assuming that you use one fine-tuned model to handle all kinds of  text-based queries rather than applying one LoRA per type of query)

2. the bar-level alignment is interesting. How is the aligned data prepared -- synthesizing the ABC into audio or applying MIR tools to audio? Could you handle audio whose tempo is unstable, both in the training and inference stages?

**Details Of Ethics Concerns:**

no concern

---

> ### Author Response · Authors · 2024-11-29
>
> Thank you for your professional and detailed review. We truly appreciate your insightful comments and have carefully considered each point you've raised.
>
> W1:  We would like to clarify that in our title, the term "Unified" was intended to highlight the integration of symbolic music and waveform modalities, rather than to imply a comprehensive unification of music and language. We understand that this phrasing may have caused some misunderstandings, and we will revise it in future iterations to improve clarity. Regarding our problem formulation, since the outputs of symbolic music generation and music understanding tasks are textual, we believe our current problem definition remains appropriate. However, we will refine our descriptions to minimize potential misinterpretations.
>
> We agree with your observation that our model can be partially described as an "audio-informed, text-based music language model." At present, our primary focus is indeed on music understanding. We acknowledge that achieving fine-grained control over waveform music remains a significant challenge, one that requires advances in semantically rich and fine-grained representation learning—an objective that forms the core idea of this work. If possible, we would be very interested in your perspective: Which approach do you think is more promising for achieving a true unification of music and language—interleaving music/audio tokens directly within the generation process (AnyGPT-style), relying on downstream tools for modality transformation (NExTGPT-style), or perhaps an entirely different approach?
>
>
> W2: We selected ChatMusician, MuPT, and Mu-LLaMA as baselines due to their architectural and setting similarities to our method. We acknowledge your suggestion to include Music Transformer and hierarchical diffusion models as baselines in future work. These models were not included initially due to their lack of support for ABC input and output formats, which made direct comparisons challenging. That said, we now realize that metrics such as FD, FAD, and KL could help address this gap, and we will incorporate these metrics in subsequent evaluations.
>
> For the Music Understanding task, we differentiate Music-Language tasks (e.g., Music Captioning, Understanding, QA) from traditional MIR tasks. The latter typically focuses on specific attributes of music, without considering natural language understanding and reasoning capabilities. Therefore, we believe our evaluation approach is justified. That said, we also see the value in your suggestion to include traditional MIR models and datasets for evaluating MuLM. Indeed, we found in our experiments that some music captioning test sets are constructed from metadata, which may lead models to overfit templates rather than genuinely understanding the music. This is an issue we will address in future work.
>
> Q1: Yes, considering this issue, we chose to train the model using LoRA to minimize the loss of the model's inherent language and instruction-following abilities. By employing a unified LoRA, we aimed to maximize complementary effects within the training set. For future revisions, we will test new types of questions and include examples to further demonstrate the model's generalization capabilities.
>
> Q2: Yes, the aligned data is prepared by synthesizing ABC notation into audio waveforms using the original tempo specified in the notation. We conduct alignment within a single bar, assuming that tempo variations are minimal in such short segments, so we have not explicitly handled unstable tempos. We will statistically analyze tempo distributions.
>
> We also recognize the limitations of our current approach, as highlighted by other reviewers. At present, the system only processes single-track music synthesized with a single instrument. While this setup ensures experimental control, it does not yet account for multi-track compositions, diverse instrumentation, or real-world scenarios involving noisy, non-synthetic music. As you suggested, leveraging MIR tools to extract ABC from real audio is a promising direction. We are excited to explore this in future work, as it could enable the model to handle real-world multi-track compositions and more diverse instrumentation more effectively.
>
> Thank you again for bringing up these important points. We believe these insights will help us improve the robustness and applicability of our method in future iterations.

---

### Meta-Review · Area_Chair_Skwg · 2024-12-21

**Metareview:**

**Paper Summary:**

This paper describes a multimodal model of symbolic music (ABC format), audio (CLAP, MERT, and Encodec embeddings), and text prompts. This is achieved by fine-tuning Llama3-8B on a variety of text+audio and text+symbolic music datasets. Experimental results claim state-of-the-art results on 5 categories of music-oriented tasks: music theory knowledge, audio captioning & QA, and symbolic generation (continuation and inpainting).

**Strengths:**

Reviewers appreciated the proposed framework for cross-modal musical reasoning, in particular, the bar-level alignment for localizing information between symbolic and audio music.

**Weaknesses:**

Reviewers raised many questions about the empirical evaluation of this model, and about the conclusions we should draw from these evaluations. Reviewer iF7b raised concerns about discrepancy between the generality of the paper’s framing and the concrete contributions: generation is limited to single-track ABC, audio features are input-only. Reviewers iF7b and N56W raised concerns about whether results are compared against the strongest possible baselines. Reviewers dR3n and RpSP note that the link between modalities is achieved using synthetic training data pairs, and raise concerns about the impact of this approach in a real-world setting.

**Additional Comments On Reviewer Discussion:**

Several of the important concerns raised by reviewers were acknowledged by the authors, but they were not adequately addressed during the discussion period. I encourage the authors to continue revising their work and incorporate this good feedback into a future version of this paper.

---

### Decision · Program_Chairs · 2025-01-22

Reject